# Cinnamyl Alcohol Attenuates Adipogenesis in 3T3-L1 Cells by Arresting the Cell Cycle

**DOI:** 10.3390/ijms25020693

**Published:** 2024-01-05

**Authors:** Yae Rim Choi, Young-Suk Kim, Min Jung Kim

**Affiliations:** 1Division of Food Functionality Research, Korea Food Research Institute, Wanju-gun 55365, Republic of Korea; uiu7896@gmail.com; 2Department of Food Science and Biotechnology, Ewha Womans University, Seoul 03760, Republic of Korea; yskim10@ewha.ac.kr

**Keywords:** cinnamyl alcohol, adipogenesis, cell cycle arrest, mitotic clonal expansion, 3T3-L1 preadipocyte

## Abstract

Cinnamyl alcohol (CA) is an aromatic compound found in several plant-based resources and has been shown to exert anti-inflammatory and anti-microbial activities. However, the anti-adipogenic mechanism of CA has not been sufficiently studied. The present study aimed to investigate the effect and mechanism of CA on the regulation of adipogenesis. As evidenced by Oil Red O staining, Western blotting, and real-time PCR (RT-PCR) analyses, CA treatment (6.25–25 μM) for 8 d significantly inhibited lipid accumulation in a concentration-dependent manner and downregulated adipogenesis-related markers (peroxisome proliferator-activated receptor γ (PPARγ), CCAAT/enhancer-binding protein α (C/EBPα), fatty acid binding protein 4 (FABP4), adiponectin, fatty acid synthase (FAS)) in 3-isobutyl-1-methylxanthine, dexamethasone, and insulin(MDI)-treated 3T3-L1 adipocytes. In particular, among the various differentiation stages, the early stage of adipogenesis was critical for the inhibitory effect of CA. Cell cycle analysis using flow cytometry and Western blotting showed that CA effectively inhibited MDI-induced initiation of mitotic clonal expansion (MCE) by arresting the cell cycle in the G_0_/G_1_ phase and downregulating the expression of C/EBPβ, C/EBPδ, and cell cycle markers (cyclin D1, cyclin-dependent kinase 6 (CDK6), cyclin E1, CDK2, and cyclin B1). Moreover, AMP-activated protein kinase α (AMPKα), acetyl-CoA carboxylase (ACC), and extracellular signal-regulated kinase 1/2 (ERK1/2), markers of upstream signaling pathways, were phosphorylated during MCE by CA. In conclusion, CA can act as an anti-adipogenic agent by inhibiting the AMPKα and ERK1/2 signaling pathways and the cell cycle and may also act as a potential therapeutic agent for obesity.

## 1. Introduction

Cinnamyl alcohol (CA), a naturally occurring aromatic compound, is found in various plant-based sources, including cinnamon leaves, Balsam of Peru, hyacinth, and storax [1,2]. Its distinct sweet, balsamic, spicy, and cinnamon-like aroma and flavor make it a sought-after ingredient in the perfume and food industry, where it is used in perfumes, scented products, baked goods, and flavored beverages [3,4]. Moreover, it serves as a precursor in industrial applications involved in the synthesis of chemicals like cinnamyl esters and taxol [5]. The derivatives of cinnamic acid can function as biologically active compounds as they possess antioxidant, anti-inflammatory, hypolipidemic, antiviral, and antimicrobial properties [6,7]. CA also has demonstrated potential medicinal properties beyond its industrial and culinary applications, including anti-inflammatory and antimicrobial activities [8,9]. Notably, it has received FDA approval for its inclusion in allergenic epicutaneous patch tests, a diagnostic tool aimed at assisting in the diagnosis of allergic contact dermatitis in individuals aged six years or older [3]. Due to its multifaceted benefits, CA continues to be of interest in various fields because of its pharmacological efficacy and versatility.

One of the popular targets is obesity, which has emerged as a global health crisis with significant implications for public health and well-being [10]. The expansion of adipose tissue through the proliferation and differentiation of preadipocytes into mature adipocytes is a key factor in obesity [11]. Adipogenic differentiation involves four key stages. The first stage of adipocyte differentiation is growth arrest [12], wherein preadipocytes exit the cell cycle and stop dividing. After growth arrest, preadipocytes enter the mitotic clonal expansion (MCE) stage [13,14]. During MCE, preadipocytes undergo multiple rounds of cell division without any increase in cell size. Growth-arrested 3T3-L1 preadipocytes can be immediately induced by hormones to express CCAAT/enhancer-binding protein β (C/EBPβ) and C/EBPδ, which are the members of the C/EBP family [15]. C/EBPβ initiates post-confluent mitosis and clonal expansion, facilitating cell cycle progression and re-entry into the cell cycle by stimulating cyclins and cyclin-dependent kinase (CDKs) in 3T3-L1 preadipocytes [16]. The cyclin-CDK complexes, such as cyclin D-CDK4/6 complexes, cyclin E-CDK2, cyclin A-CDK2, and cyclin B-CDK1, are activated and regulated during the G_1_ phase, the G_1_/S checkpoint, the S phase, and the G_2_/M phase, respectively [17]. MCE is followed by the intermediate differentiation stage. During this stage, preadipocytes begin to acquire certain characteristics of mature adipocytes but are not fully differentiated yet. The final step in adipocyte differentiation is terminal differentiation. During this stage, the maturation of preadipocytes into adipocytes is completed, characterized by the synthesis and accumulation of lipid droplets, the hallmark feature of fat cells [18]. C/EBPβ and C/EBPδ induce the expression of the master transcription regulator genes involved in adipogenesis, such as proliferator activated receptor γ (PPARγ) and C/EBPα genes [19]. These four stages of adipocyte differentiation are regulated by a complex network of molecular events. Dysregulation in any of these stages can disrupt the process and may be associated with metabolic disorders such as obesity. Understanding the molecular mechanisms underlying each stage of adipocyte differentiation is crucial for advancing our knowledge of adipogenesis and its implications for overall metabolic health.

In the adipogenesis process of 3T3-L1 cells, adenosine monophosphate (AMP)-activated protein kinase (AMPK) and the phosphoinositide 3-kinase/mitogen-activated protein kinase (PI3K/MAPK) pathways play crucial roles. AMPK is a serine/threonine protein kinase that regulates cellular energy homeostasis [20]. Under energy depletion and metabolic stress, AMPK inhibits the ATP-consuming metabolic reaction, which leads to an increase in the intracellular AMP/ATP ratio and stimulates ATP-generating reactions to maintain the normal homeostasis of cellular energy [21]. AMPK phosphorylation promotes the biosynthetic pathways and regulates fatty acid oxidation to enhance energy expenditure. Therefore, AMPK is a potential therapeutic target of the anti-adipogenic factor, which activates the modulation of various metabolic processes, such as the regulation of obesity, and serves a crucial role in lipid homeostasis. On the other hand, the PI3K/MAPK pathway, vital for cell growth, division, and survival, positively regulates adipogenesis through its activation, which promotes cell cycle progression and differentiation. Consequently, AMPK and PI3K/MAPK pathways perform complementary or opposing functions in adipocyte formation and function, establishing them as significant factors in the regulation of adipocyte development and activity.

To our knowledge, only one study has reported the inhibitory effect of CA on adipocyte differentiation [22]; however, only a few adipogenic transcription factors in connection to the mechanism involved were examined. To ascertain the treatment time for CA, a potential anti-obesity ingredient, it is necessary to know the exact target time. Therefore, an exhaustive examination of the mechanism through which CA affects obesity is necessary. This study aimed to investigate the effects of CA on adipogenesis in 3T3-L1 preadipocytes and the underlying mechanisms. We identified the stages of adipogenesis affected by CA and assessed changes in related biomarkers to elucidate the mechanism through which CA targets obesity.

## 2. Results

### 2.1. Effect of CA on Cytotoxicity and Lipid Accumulation in 3T3-L1 Adipocytes

Before examining the effect of CA (Figure 1A) on the differentiation of 3T3-L1, an experiment was conducted to establish the treatment concentration. The cytotoxicity of CA was measured at concentrations of 0–100 μM, and it was found that at concentrations ≤ 25 μM, CA exhibited no cytotoxicity in 3T3-L1 cells (Figure 1B). Therefore, CA was used at the concentration range of 6.25–25 μM. Lipid droplets in 3T3-L1 cells were visualized and quantified using Oil Red O (ORO) staining (Figure 1C). The mixture of 0.5 mM of IBMX, 1 μM of DEX, and 1 μg/mL of insulin (MDI) successfully induced adipogenesis and lipid accumulation, whereas CA treatment significantly inhibited MDI-induced lipid accumulation in a concentration-dependent manner. The protein and mRNA expression levels of adipogenesis-related biomarkers were also regulated by CA. While MDI treatment upregulated PPARγ, C/EBPα, and FABP4 expression, CA treatment inhibited the MDI-induced upregulation of these proteins (Figure 1D). In addition, CA significantly suppressed the MDI-induced upregulation of the mRNA expressions of *Pparg*, *Cebpα*, *Fabp4*, *Adipoq*, and *Fasn* (Figure 1E).

### 2.2. Regulation of Adipogenesis by CA at the Early Phases

To investigate the mechanism of the anti-adipogenesis effect of CA, 3T3-L1 preadipocytes were treated with CA at various time points throughout differentiation (Figure 2A). As assessed via ORO staining, CA (25 μM) attenuated MDI-induced lipid accumulation in 3T3-L1 cells treated for 0–2, 0–4, and 0–8 d compared to the differentiation (DIF) group but did not influence cells treated for 2–4 and 4–8 d (Figure 2B). Therefore, the first two days were critical for CA to inhibit intracellular lipid accumulation. Consistent with Figure 2B, the protein expression levels of PPARγ, C/EBPα, and FABP4 were significantly suppressed by CA treatment of MDI-treated 3T3-L1 cells compared to the DIF group (Figure 2C). This indicated that CA regulated the early phases of adipocyte differentiation.

### 2.3. Inhibition of Cell Cycle Progression during MCE by CA

We hypothesized that the anti-adipogenic activity of CA was closely involved with the cell cycle because MCE occurs during the early stages of preadipocyte differentiation, and cell cycle progression plays a role in MCE. The number of 3T3-L1 cells increased after MDI treatment (the control group) but not after CA treatment for 24 and 48 h (Figure 3A). Flow cytometric analysis showed that MDI induced cell cycle progression from the G_0_/G_1_ phase to the S and G_2_/M phases, which was attenuated by CA (Figure 3B). After 16 h of cell cycle progression, the percentage of cells in the G_0_/G_1_ phase changed from 82.83% to 45.21% and 65.45% in the control and CA-treated groups, respectively, and the percentage of cells in the S phase changed from 3.58% to 37.72% and 19.53% in the control and CA-treated groups, respectively. Biomarkers related to transcription factors (C/EBPβ and C/EBPδ) and the cell cycle (cyclin D1 and CDK6 for the G_1_ phase, cyclin E1 and CDK2 for the G_1_/S checkpoint, and cyclin B1 for G_2_/M phase) were also regulated by CA (Figure 3C,D). MDI raised the protein levels of C/EBPβ, C/EBPδ, cyclin D1, CDK6, cyclin E1, CDK2, and cyclin B1 in 3T3-L1 preadipocytes, while CA significantly attenuated their expression levels at 16 and 20 h. These results indicate that CA inhibits the cell cycle by trapping cells in the G_0_/G_1_ phase at the onset of adipogenesis.

### 2.4. Effect of CA on the AMPKα and ERK1/2 Signaling Pathways

Two upstream signaling pathways involved in adipogenesis were also investigated: the AMPK pathway with its downstream targets and the PI3K/MAPK pathway. The biomarkers investigated within the AMPK pathway include AMPK and its downstream component, acetyl-CoA carboxylase (ACC). Typically, AMPK is activated through phosphorylation, which further leads to the phosphorylation and inactivation of ACC. Compared to the MDI-treated group, CA effectively increased the phosphorylation of both AMPKα and ACC, especially at 30 min after MDI treatment (Figure 4A). Assessment of the PI3K/MAPK signaling pathway revealed that CA effectively suppressed the phosphorylation of AKT and extracellular signal-regulated kinase 1/2 (ERK1/2) at 15 and 30 min, respectively. However, CA did not exhibit any inhibitory effect on the phosphorylation of p38 MAPK (Figure 4B).

## 3. Discussion

The accumulation of excessive lipids in adipose tissue is a characteristic of obesity, and the effectiveness of anti-obesity interventions frequently hinges on their capacity to restrict lipid buildup [10,23]. Therefore, in vitro experiments on anti-obesity effects often seek to determine the anti-adipogenic potential of a target compound. Several studies have examined the effects of natural compounds on adipocyte differentiation and related gene regulation [24]. In this study, we investigated the anti-adipogenic effect of CA and the associated regulatory mechanism. Adipocyte differentiation is a complex process regulated by cell morphology, various transcription factors, and adipogenesis-related genes [25]. C/EBPβ is expressed in the early stages of adipocyte differentiation and activates C/EBPα and PPARγ, which are expressed during the terminal differentiation of adipocytes [19,26,27]. During adipogenesis, C/EBPα and PPARγ activate the expression of lipid metabolism enzymes such as FABP4, aP2, and FAS [26,28]. Our results showed that CA inhibited lipid accumulation at concentrations of 6.25–25 µM and downregulated the expression of *Pparg*, *Cebpα*, *Fabp4*, *Adipoq*, and *Fasn* at 25 µM. This concentration was lower than the previously reported effective concentrations of 5 and 10 µg/mL (37 and 74 µM, respectively) that inhibited the expression of PPARγ, C/EBPα, SREBP-1c, and FAS [22]. In previous studies, cinnamon extract containing CA and other components of cinnamon is also effective as an anti-obesity agent [29]. Cinnamaldehyde, the main component of cinnamon, inhibits lipid accumulation and the expression of PPARγ, C/EBPα, and SREBP-1 in 3T3-L1 adipocytes [30]. Furthermore, the treatment of high-fat-induced mice with cinnamaldehyde led to weight loss; decreased plasma lipid levels; and suppressed the expression of adipogenesis-related target genes, such as the *Fasn*, *Fapb4*, and *Lpl* genes, in epididymal fat. Cinnamyl isobutyrate, a cinnamon-derived bioactive aromatic compound, reduces lipid accumulation and protein expression levels of PPARγ, C/EBPα, and FABP4 after treatment for 12 d [31]. Hydroxylated cinnamoyl ester, particularly 3,4,5-trihydroxycinnamic acid decyl ester, suppresses triglyceride levels and lipid accumulation during differentiation [32].

We found that CA regulates MCE, the early stage of differentiation. Differentiation of 3T3-L1 preadipocytes into adipocytes occurs in several stages and includes initial (MCE), intermediate, and terminal differentiation stages [33]. In the MCE stage, hormones induce the growth of quiescent confluent preadipocytes, which replicate one to two times and double their cell number [12]. In the intermediate stage (on day 4 of differentiation), cytoplasmic lipid droplet formation starts, and lipid accumulation in the cells becomes visible. In the terminal stage, the cells are completely differentiated into mature adipocytes. As MCE is an essential step in adipocyte differentiation, inhibiting MCE alone can inhibit differentiation. Curcumin, ginsenoside CK, and lactucin inhibit adipocyte differentiation by regulating the early stage (days 0–2) [34,35,36]. Apigenin, which is abundant in common fruits and vegetables, inhibits adipogenesis by suppressing the formation of lipid droplets in the early stages of adipocyte differentiation [37]. Likewise, our results showed that CA inhibits adipogenesis after treatment during the early stage of differentiation (MCE stage; 0–2 d), similar to treatment for 0 to 8 days (Figure 2).

In MCE, C/EBPβ, an important transcription factor in the early stages of adipogenesis, and cell cycle progression are essential for the terminal differentiation of 3T3-L1 adipocytes [19]. The cell cycle, consisting of the interphase and mitotic phases, is intricately regulated by specific cyclins and CDKs at each phase [38,39]. During interphase, which encompasses the G_1_, S, and G_2_ stages, different cyclin-CDK complexes control the cell cycle progression. In the G_1_ phase, cyclin D partners with CDK4/6 to initiate the cell cycle [40]. Cyclin E interacts with CDK2 to induce the cell cycle from the G1 to the S phase. In the S phase, cyclin A binds to CDK2 to facilitate DNA replication. Finally, in G_2_, cyclin A and CDK1 prepare the cell for mitosis. In the mitotic phase (M phase), cyclin B works with CDK1 to orchestrate cell division. These specific partnerships ensure that the cell cycle proceeds smoothly and accurately, allowing for proper replication and division while safeguarding cellular functionality. Therefore, inhibiting the activation of C/EBPβ and arresting the cell cycle during the MCE stage suppresses differentiation [17]. The inhibitory effect of apigenin occurs by regulating the DNA-binding activity of C/EBPβ and inhibiting the G_0_/G_1_ stage during the cell cycle in MCE [37]. Curcumin, a phenolic compound found in turmeric, delays the G_1_/S phase transition in adipogenic differentiation by inhibiting CDKs, reducing Rb phosphorylation and cyclin D1 expression, and increasing p27Kip1 expression, thereby suppressing MCE and adipocyte differentiation [34]. We confirmed that CA inhibits C/EBPβ and C/EBPδ, maintains cells in the G1 phase of the cell cycle, and delays the entry of preadipocytes into the S phase and mitosis by downregulating the expression of cyclin D1 and CDK6, thereby delaying adipocyte differentiation (Figure 3).

Several upstream signaling pathways of C/EBPβ are involved in the early stages of 3T3-L1 adipocyte differentiation, and AMPK is one of them. AMPKα plays a crucial role in regulating various biological pathways, such as lipid metabolism and cellular energy homeostasis [33,41]. As an energy sensor within the cell, AMPKα modulates the activity of enzymes involved in ATP production and consumption. During adipocyte differentiation, AMPKα activation leads to the phosphorylation of ACC and its conversion to malonyl-CoA, which further activates carnitine palmitoyltransferase-1 (CPT-1) and promotes β-oxidation, ultimately increasing the transport of fatty acids into the mitochondria [33]. This process inhibits fatty acid synthesis and activates signaling pathways that promote fatty acid degradation, suppressing adipocyte differentiation and potentially reducing fat accumulation in the process. Cinnamon extract activates AMPK-ACC signaling to inhibit lipogenesis and promote glucose uptake [42]. Cinnamaldehyde inhibits lipid droplet formation by phosphorylating AMPK and ACC [30]. Although cinnamaldehyde decreases the expression of the *SREBP-1* and *FAS* genes, which are downstream of AMPK, PPARγ and C/EBPα are not downstream signals of AMPK. Similarly, CA activates AMPK and inhibits ACC by phosphorylating it.

The PI3K/AKT and MAPK/ERK signaling pathways play a significant role in the early stages of adipocyte differentiation, particularly in the induction of MCE. This hormonal cocktail activates the pathways involved in the induction of MCE, wherein PI3K/AKT signaling and intracellular MAPK signaling are crucial for regulating cellular processes, including the initiation of the cell cycle, proliferation, and differentiation. MAPK/ERK activation phosphorylates C/EBPβ and plays an important role in MCE and adipogenesis [43]. Inactivation of the PI3K/AKT pathway in 3T3-L1 preadipocytes inhibits adipogenesis, while its activation contributes to adipocyte differentiation [44]. Cinnamon extract regulates the phosphorylation of AKT, which acts downstream of the insulin receptor through a pathway different from the AMPK pathway [42]. Sulforaphane, an isothiocyanate compound, decreases the phosphorylation of ERK1/2 and AKT in the early stages [45]. Glycyrrhizic acid, a major component of licorice root, inhibits the early stage of adipogenesis by suppressing MEK/ERK-mediated C/EBPβ and C/EBPδ expression [46]. Likewise, CA inhibits adipogenesis by activating the phosphorylation of AMPKα and downregulating the ERK1/2 signaling pathway (Figure 4).

In this study, we successfully found that CA inhibits adipogenesis in 3T3-L1 cells by interfering with entry into MCE during the early stage of differentiation through the AMPK pathway and MAPK/ERK pathway. However, this study also has limitations, the biggest of which is that it was tested only in an in vitro model. While in vitro models are recognized as a useful preliminary screening step to evaluate the mechanism of natural anti-obesity agents, they constitute a major limitation in this study. Therefore, further studies using animal and human experiments under obesity-induced conditions are required. Nevertheless, the CA may serve as a potential anti-obesity agent.

## 4. Materials and Methods

### 4.1. Reagents

Dexamethasone (DEX), 3-isobutyl-1-methylxanthine (IBMX), insulin, ORO, isopropyl alcohol, and CA were obtained from Sigma-Aldrich (St. Louis, MO, USA). All antibodies against C/EBPβ, C/EBPδ, PPARγ, C/EBPα, FABP4, cyclin D1, CDK6, cyclin E1, CDK2, cyclin B1, AMPKα, phosphorylated-AMPKα, ACC, phosphorylated-ACC, β-actin, anti-rabbit IgG, and anti-mouse IgG HRP-linked antibody were procured from Cell Signaling Technology (Danvers, MA, USA). Dulbecco’s Modified Eagle’s medium (DMEM), bovine calf serum (BCS), fetal bovine serum (FBS), and penicillin/streptomycin (P/S) were obtained from Gibco (Grand Island, NY, USA).

### 4.2. Cell Culture and Differentiation

Briefly, 3T3-L1 preadipocytes were obtained from the American Type Culture Collection (ATCC, Manassas, VA, USA) and grown in DMEM medium containing 10% BCS and 1% P/S. After attaining 100% confluency, the culture was incubated for an additional 2 days, following which the medium was changed (day 0) to DMEM + 10% FBS containing a differentiation cocktail (MDI; the mixture of 0.5 mM IBMX, 1 μM DEX, and 1 μg/mL insulin) that was removed 48 h later (day 2). Next, 3T3-L1 cells were cultured for another four days in a modified medium containing 1 μg/mL of insulin in DMEM+10% FBS. The medium was changed every 2 days.

### 4.3. Cytotoxicity

The viability of 3T3-L1 cells was measured using a Cell Counting Kit-8 (CCK-8) kit (Dojindo, Kumamoto, Japan). 3T3-L1 preadipocytes were seeded in a 96-well plate at a density of 50,000 cells/well. The next day, cells were treated with CA (6.25–100 μM) for 24 and 48 h. Subsequently, CCK-8 solution was added to each well and incubated for 2 h. Absorbance was measured at 450 nm using a microplate reader (SpectraMax M2, Molecular Devices, Sunnyvale, CA, USA).

### 4.4. Trypan Blue Staining

Cell number was measured using the trypan blue assay. 3T3-L1 preadipocytes were seeded in a 6-well plate at a density of 50,000 cells/well. Next day, preadipocytes were cultured in an MDI-containing medium in the presence or absence of CA for 16 and 20 h. Then, the cells were trypsinized and resuspended in a medium containing 0.4% trypsin blue solution (1:1). The number of viable cells was measured using a LUNA-II cell counter (Logo Biosystems, Gyeonggi-do, Republic of Korea).

### 4.5. Oil Red O (ORO) Staining

3T3-L1 preadipocytes were cultured in a 12-well plate, and their differentiation into mature adipocytes was induced. The culture was incubated for an additional 2 days, following which the medium was changed (day 0) to DMEM+10% FBS containing MDI that was removed 48 h later (day 2). Next, 3T3-L1 cells were cultured for another four days in a modified medium containing 1 μg/mL of insulin in DMEM+10% FBS. On day 8, the cells were washed with phosphate-buffered saline (PBS) and fixed in 4% paraformaldehyde for 15 min. After rinsing with PBS, the cells were treated with 60% isopropanol for 5 min and stained using 0.5% ORO solution for 10 min. The cells were further washed twice with PBS. Microscopic images of stained lipid droplets were captured and quantified by eluting them with 100% isopropanol. Absorbance was measured at 500 nm using a microplate reader.

### 4.6. Western Blot

Under MDI treatment, CA-treated and untreated cells were lysed with the RIPA buffer containing protease inhibitors and centrifuged (13,500 rpm) for 30 min at 4 °C. The total protein concentration was measured using a bicinchoninic acid protein assay. The quantified proteins were denatured by heating them at 105 °C for 5 min and further separated using sodium dodecyl sulfatepolyacrylamide gel electrophoresis (SDS-PAGE) and transferred onto polyvinylidene difluoride (PVDF) membranes. The membranes were blocked with 5% bovine serum albumin (BSA) and incubated with primary antibodies at 4 °C overnight. Subsequently, the membranes were washed and incubated with secondary antibodies for 2 h. The protein bands were detected using enhanced chemiluminescence.

### 4.7. Real-Time Polymerase Chain Reaction (RT-PCR)

Under MDI treatment, CA-treated and untreated total RNA were extracted using TRIzol (Thermo Fisher Scientific, Waltham, MA, USA), following the manufacturer’s instructions. Isolated RNA (1 µg) was used for cDNA synthesis using M-MulV reverse transcriptase, dNTPs, and an oligo (dT) 18 primer. RT-PCR was used to amplify cDNAs using the SYBR green master mix (Applied Biosystems, Foster City, CA, USA). The RT-PCR included 40 cycles of denaturation at 95 °C for 15 s, annealing at 60 °C for 60 s, and extension at 60 °C for 30 s. The target primer sequences used in RT-PCR were as follows: *Pparg* forward 5′-AAGGATTCATGACCAGGGAGTTCC-3′ and reverse 5′ R: GCGGTCTCCACTGAGAATAATG-3′; *Cebpα* forward 5′-GTGGACAAGAACAGCAACGAGT-3′ and reverse 5′-AGGCGGTCATTGTCACTGGTCAA-3′; *Fapb4* forward 5′-GTGGGCTTTGCCACAAGGAAAGT-3′ and reverse 5′-GGTGATTTCATCGAATTCCACGCC-3′; *Adipoq* forward 5′-AGCCGCTTATGTGTATCGCTCAG-3′ and reverse 5′- CCCGGAATGTTGCAGTAGAACT-3′; *Fasn* forward 5′-TGGGTTTGGTGAATTGTCTCCG-3′ and reverse 5′-ACACGTTCATCACGAGGTCATG-3′; *Gapdh* forward 5′-AATGTGTCCGTCGTGGATCTGA-3′ and reverse 5′-TTGCTGTTGAAGTCGCAGGAGA-3′. The mRNA levels of all genes were normalized using *Gapdh*.

### 4.8. Cell Cycle Analysis

Differentiation of 3T3-L1 cells was induced by MDI, followed by treatment with CA. The cells were harvested and washed with PBS. Next, the cells were fixed and permeabilized using 70% ethanol at −20 °C overnight. The fixed cells were harvested and washed twice. Next, the cells were stained with 50 μg/mL of propidium iodide (PI) solution containing 50 μg/mL of RNase A in PBS. Cell cycle analysis was performed using CytoFLEX (Beckman Coulter, Indianapolis, IN, USA), and data analysis was performed using CytExpert 2.4 software (Beckman Coulter, Indianapolis, IN, USA) to determine the relative cell numbers.

### 4.9. Statistical Analysis

All data, expressed as mean ± standard deviation (*n* = 3), were analyzed using GraphPad Prism 9.0 (GraphPad Software, San Diego, CA, USA). One-way and two-way analyses of variance were performed with Dunnett’s and Tukey’s multiple comparison tests to identify significant differences between group means. A *p*-value < 0.05 was considered statistically significant.

## 5. Conclusions

Although the efficacy of CA in suppressing adipogenic differentiation has been previously reported, we confirmed its anti-adipogenic effects at lower concentrations. Moreover, to our knowledge, this study is the first to elucidate the mechanism through which CA suppresses adipogenesis by regulating MCE, which is an early stage in adipocyte differentiation. CA inhibited the adipogenesis of 3T3-L1 cells induced by the hormonal cocktail MDI and downregulated the expression of adipogenesis-related transcription factors, such as PPARγ, C/EBPα, FABP4, adiponectin, and Fas. In particular, CA regulated the early stage of adipocyte differentiation. In MCE, CA attenuated the levels of C/EBPβ and C/EBPδ and delayed the differentiation of preadipocytes into adipocytes by arresting the cell cycle. This was confirmed by changes in cell cycle markers, CDKs, and cyclins. Moreover, at the early stages of differentiation, CA regulated the upstream signaling pathways by activating AMPK phosphorylation and inhibiting ERK1/2 phosphorylation. This study has demonstrated the anti-adipogenic effect of CA on 3T3-L1 cells and elucidated the underlying mechanism, validating its potential as an anti-obesity agent.

## Figures and Tables

**Figure 1 ijms-25-00693-f001:**
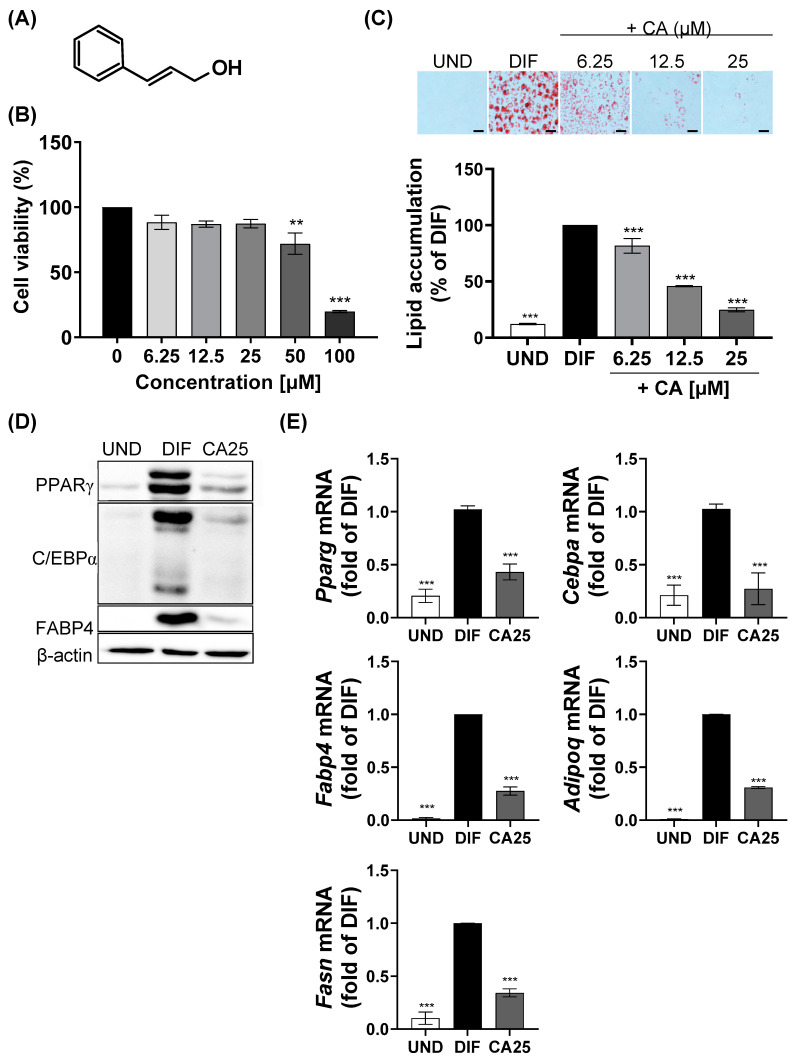
Effect of CA on 3T3-L1 cells. (**A**) Structure of CA. (**B**) Cytotoxicity of CA on 3T3-L1 preadipocytes at 48 h. (**C**) Changes in lipid accumulation on 3T3-L1 adipocytes treated with CA during differentiation (scale bars, 40 μm). (**D**) Protein expression of PPARγ, C/EBPα, and FABP4. (**E**) Gene expression levels of *Pparg*, *Cebpa*, *Fabp4*, *Adipoq*, and *Fasn*. ** *p* < 0.01, *** *p* < 0.001 vs. control or DIF group. All experiments were performed in triplicate. UND, undifferentiated 3T3-L1 cells; DIF, differentiated 3T3-L1 cells; CA25, cinnamyl alcohol (25 µM); PPARγ, peroxisome proliferator-activated γ; C/EBPα, CCAAT/enhancer-binding protein α; FABP4, fatty acid binding protein 4; Fasn, fatty acid synthase; Adipon, adiponectin.

**Figure 2 ijms-25-00693-f002:**
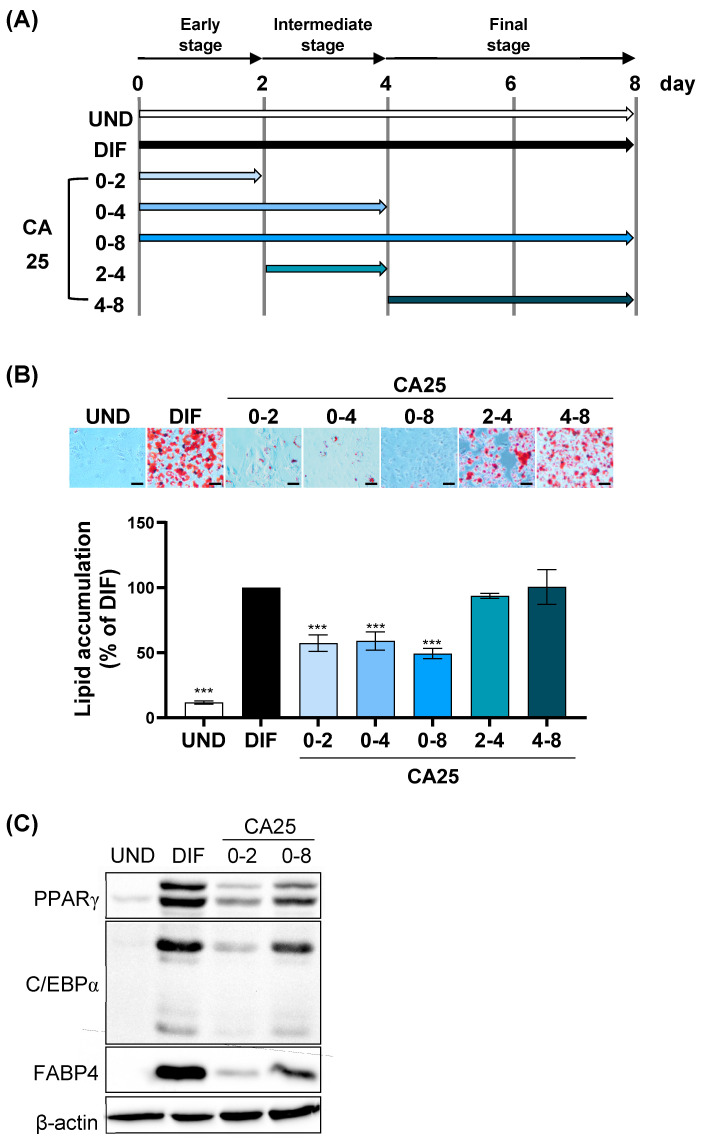
Effect of CA on the differentiation of 3T3-L1 cells at different stages. (**A**) Schematic representation of various exposure periods during differentiation. (**B**) Changes in lipid accumulation by CA at different stages (scale bars, 40 μm). (**C**) Protein expression of PPARγ, C/EBPα, and FABP4. *** *p* < 0.001 vs. DIF group. All experiments were performed in triplicate. UND, undifferentiated 3T3-L1 cells; DIF, differentiated 3T3-L1 cells; CA25, cinnamyl alcohol (25 µM); PPARγ, peroxisome proliferator-activated γ; C/EBPα, CCAAT/enhancer-binding protein α; FABP4, fatty acid binding protein 4.

**Figure 3 ijms-25-00693-f003:**
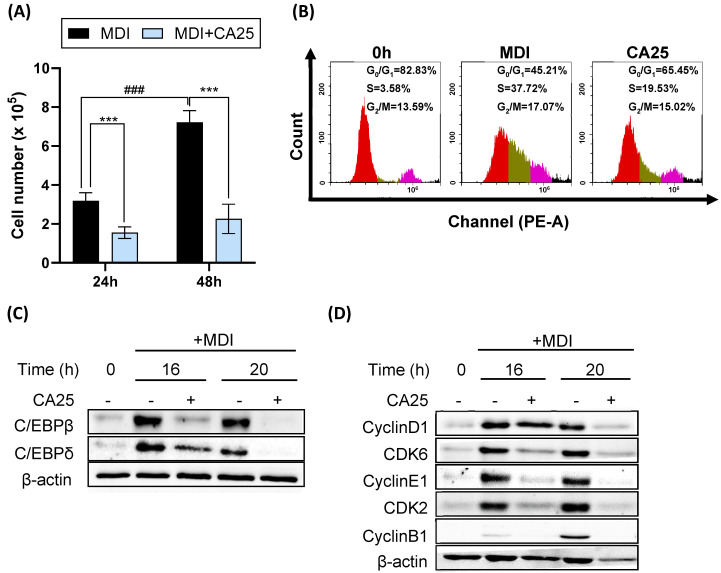
Effect of CA on MCE in 3T3-L1 preadipocytes. (**A**) CA-induced changes in total cell number at 24 and 48 h. *** *p* < 0.001, significant difference compared to the control group (CA-untreated group); ^###^
*p* < 0.001, compared to the MDI-treated group for 24 h. (**B**) Flow cytometry analysis of cell cycle distribution on 3T3-L1 preadipocytes after CA treatment for 16 h. Red, G_0_/G_1_ phase; Olive, S phase; Pink, G_2_/M phase. (**C**) Protein expression levels of C/EBPβ and C/EBPδ. (**D**) Protein expression levels of cell-cycle markers (cyclin D1, CDK6, cyclin E1, CDK2, and cyclin B1). All experiments were performed in triplicate. MDI, hormonal cocktail-treated group; CA25, cinnamyl alcohol (25 μM); C/EBPβ, CCAAT/enhancer-binding protein β; C/EBPδ, CCAAT/enhancer-binding protein δ.

**Figure 4 ijms-25-00693-f004:**
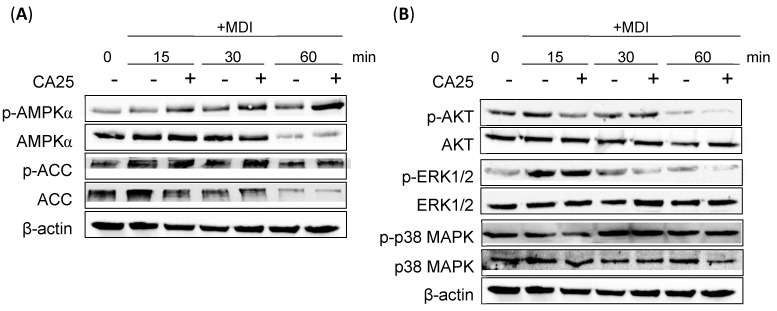
Effects of CA on the AMPKα and PI3K/MAPK signaling pathways. (**A**) Protein expression levels of AMPKα and ACC after CA treatment. (**B**) Protein expression levels of proliferation markers (ERK1/2, AKT, and p38 MAPK) after CA treatment. All experiments were performed in triplicate. MDI, hormonal cocktail-treated group; CA25, cinnamyl alcohol (25 μM); AMPKα, AMP-activated protein kinase α; ACC, acetyl-CoA carboxylase; ERK1/2, extracellular signal-regulated kinase1/2; p38 MAPK, p38 mitogen-activated protein kinase.

## Data Availability

The data presented in this study are available on request from the corresponding author.

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
