# Peer review of "Cinnamyl Alcohol Attenuates Adipogenesis in 3T3-L1 Cells by Arresting the Cell Cycle"

_ijms, 2024, doi:10.3390/ijms25020693_

Round 1

Reviewer 1 Report

Comments and Suggestions for Authors

Reviewer 2 Report

Comments and Suggestions for Authors

The authors explored influence of cinnamyl alcohol (CA) on the ability of 3T3-L1 cells to differentiate into adipocytes. The study included exploration of CA cytotoxicity, through which CA working concentration equal to 25 uM has been chosen. After that, authors estimated the ability of adipocytes to store lipids through ORO staining, expression of the key genes, related to adipogenesis, and their corresponding proteins through Western-blotting. Further on authors demonstrated that the suppression of differentiation takes place at the early stages (0-2 days of in vitro culture). Authors also assessed the phase of cell cycle at which the arrest was initiated and involved proteins and markers, as well as explored influence of CA in concentration 25 uM on the upstream signaling pathways, AMPK and ERK.-dependent.  

The results of the work may be used in the future for search of the new approaches to influence adipogenesis.

The article is interesting and well-written.

There are only minor corrections that can be done:

1.       Line 73: AMPK… stimulates ATP-generating reactions to DECREASE the intracellular AMP/ATP ratio or to INCREASE ATP/AMP ratio…

2.       Please, decipher MDI in results

3.       Abbreviations of the names of genes and proteins will enhance understanding of the figures

4.       Numbers of the performed experiments have to be indicated in the figure legends

5.       The majority of the references are old. Only 10 are related to the recent 5 years. Please, update literature sources.
